# UniAR: A Unified model for predicting human Attention and Responses on visual content

Peizhao Li* [†1,2], Junfeng He* [‡1], Gang Li* [‡1], Rachit Bhargava[1], Shaolei Shen[1], Nachiappan Valliappan[1], Youwei Liang[†1,3], Hongxiang Gu[1], Venky Ramachandran[1], Golnaz Farhadi[1], Yang Li[1], Kai J Kohlhoff[1], and Vidhya Navalpakkam[1]

[1]Google Research
[2]Brandeis University
[3]University of California San Diego

## Abstract

Progress in human behavior modeling involves understanding both implicit, early-stage perceptual behavior, such as human attention, and explicit, later-stage behavior, such as subjective preferences or likes. Yet most prior research has focused on modeling implicit and explicit human behavior in isolation; and often limited to a specific type of visual content. We propose UniAR – a unified model of human attention and preference behavior across diverse visual content. UniAR leverages a multimodal transformer to predict subjective feedback, such as satisfaction or aesthetic quality, along with the underlying human attention or interaction heatmaps and viewing order. We train UniAR on diverse public datasets spanning natural images, webpages, and graphic designs, and achieve SOTA performance on multiple benchmarks across various image domains and behavior modeling tasks. Potential applications include providing instant feedback on the effectiveness of UIs/visual content, and enabling designers and content-creation models to optimize their creation for human-centric improvements.

## 1 Introduction

Implicit, early-stage perceptual behavior such as human attention is intricately linked with explicit, later-stage behavior such as subjective ratings/preferences. Yet prior research has often studied these in isolation. For example, there is a large body of work on predictive models of human attention that are known to be useful for various applications, ranging from basic attention/eye-movement research [32, 35], to optimizing interaction designs [4, 65, 9], enhancing webpage layouts [67, 85, 11], improving user experience in immersive environments [5] and improving natural image and photo quality by reducing visual distraction [1]. Prior research has also explored predicting other kinds of implicit human behavior such as the sequence/order in which items are viewed (attention scanpath) in natural images or webpages [22, 19], assessing visual importance in graphic designs [47, 63, 27], and understanding visual clutter [52, 79, 71].

Separately from implicit, early-perceptual behavior, there has also been research in modeling explicit, later-stage decision-making behavior such as subjective preferences [20] and aesthetic quality [37, 21, 55, 30]. Prior research has been further fragmented due to dedicated models focusing on specific combinations of behavior tasks, input domain (e.g., natural images, designs, and webpages), and task scenarios (e.g., free viewing, object searching, and question answering).

---

*Co-first authors, equal technical contribution

†Work done during an internship at Google Research

‡Corresponding authors, equal leading contribution: {junfenghe,leebird}@google.com

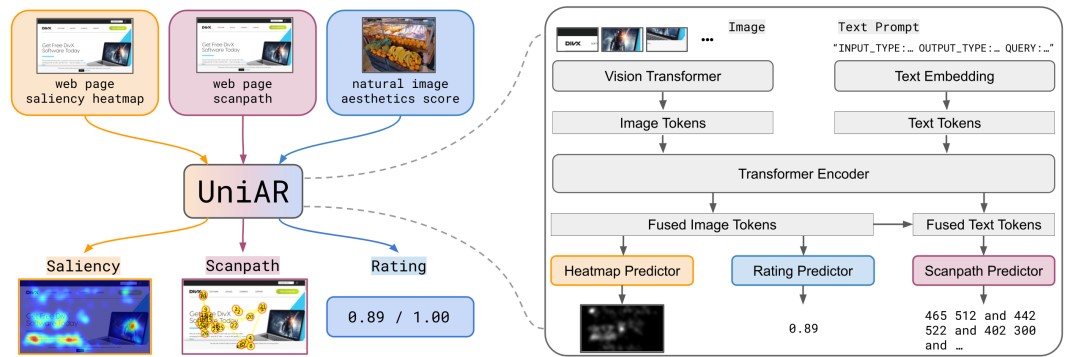

Figure 1: Overview of our UniAR model. UniAR is a multimodal model that takes an image (could be a natural image, screenshot of a webpage, graphic design, or UI) along with a text prompt as input, and outputs heatmaps of human attention/interaction, scanpath or sequence of viewing/interaction, and subjective preference/likes. Example inputs and corresponding outputs for saliency, scanpath, and rating are shown on the left side, and the detailed model architecture is shown on the right side.

To the best of our knowledge, a unified approach is still missing to modeling human visual behavior, ranging from implicit, early-perceptual behavior of what draws human attention, to explicit, later-stage decision-making on subjective preferences or likes.

In this paper, we ask the following question: *Can we build a unified model of human attention and preference behavior that reliably works across diverse types of visual content? If so, how does it compare with state-of-the-art (SOTA) models dedicated to specific domains and tasks?* Such a unified model could enable a wide variety of applications. For instance, it could augment human decision making and accelerate evaluation of effective UIs by not only predicting preferences as rewards, but also providing additional insights in the form of predicted human attention behavior.

**UniAR.** In this paper, we consider 11 public datasets consisting of different input domains/visual content (e.g., natural images, cartoons, art, graphic designs and webpages), behavior tasks (e.g., attention heatmaps, scanpath, likes/preference), and task-scenarios (e.g., free-viewing, object search, question answering). We introduce a new model, **UniAR** – A **Uni**fied model for predicting human **A**ttention and **R**esponses on visual content. UniAR is a multimodal transformer model that takes images and text prompts as input. The text prompt combines information about the input domain (e.g., natural image, graphics design or webpage), the desired behavior prediction task (e.g., attention heatmap or aesthetic score), and specifics of the task scenario when relevant (e.g., object name in an object search task). Our model generates predictions conditionally on these inputs. Experiments show that UniAR achieves SOTA performance across diverse datasets, spanning different input domains, behavior prediction tasks, and task scenarios.

**Main contributions** of this work are summarized below:

1. We proposed UniAR, a multimodal transformer model to predict different types of human behavior from attention to likes, across diverse types of visual content.

2. We trained UniAR on 11 benchmark datasets with different input domains (natural images, webpages, and graphic designs) and output behavior types (attention/importance heatmaps, viewing sequence or scanpath, and aesthetics/quality scores), and showed that UniAR, which is a single unified model, can outperform or perform comparably to SOTA models trained on specific tasks and datasets. We further showed that UniAR generalizes well to tasks with unseen input and output combinations, under a zero-shot setting.

We present various visualization results from UniAR on saliency/importance heatmap, scanpath, and ratings in Figure 2, compared to ground truth.

## 2   Related Work

**Saliency prediction.** Saliency or attention heatmap prediction is a common implicit behavioral task aimed at predicting which areas within an image are more likely to draw human attention. Saliency models can be helpful for understanding human visual attention, and have been used for applications such as evaluating the quality of UIs, optimizing content placement in graphic designs, and improving

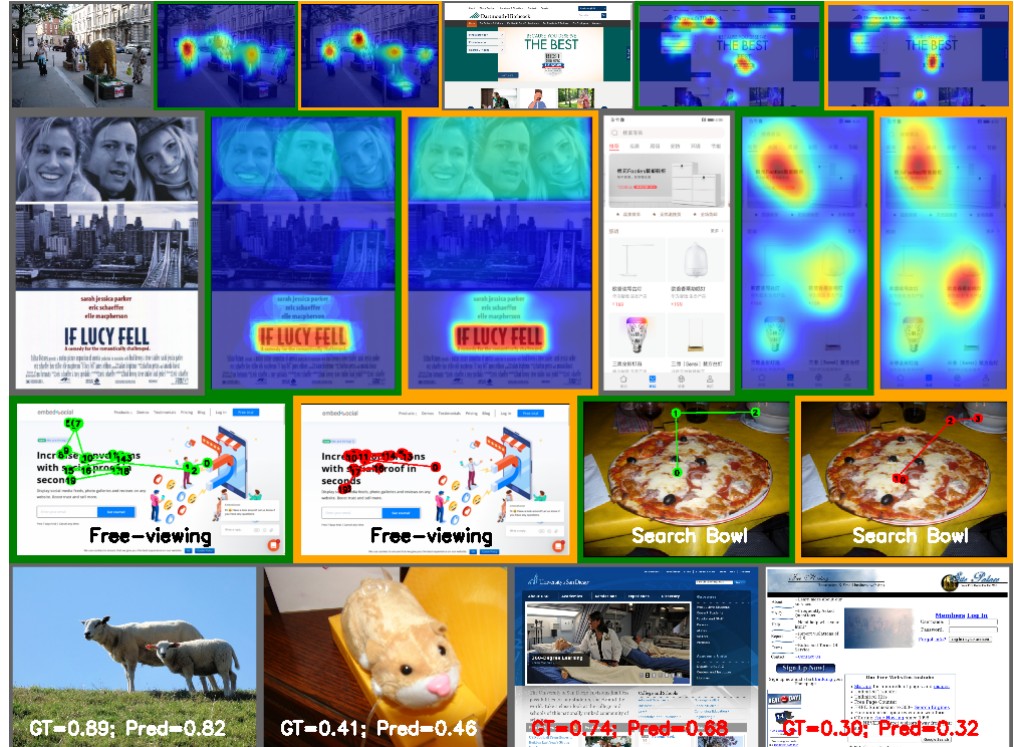

Figure 2: Examples of UniAR's predictions across different tasks/domains. Images in green border are ground-truth, while images in orange border are UniAR's predictions. **First row**: attention/saliency heatmap prediction on natural images (*Salicon*) and webpages (*WS-Saliency*). **Second row**: importance heatmap on graphic designs (*Imp1k*), and saliency heatmap on *Mobile UI*. **Third row**: scanpath-sequence during free-viewing of webpages (*WS-Scanpath*) and object-searching within images (*COCO-Search18*). **Fourth row**: preference/rating prediction for natural images (*Koniq-10k*) and webpages (*Web Aesthetics*).

perceptual quality of compressed images/videos (i.e., allocating more resources to visually important regions, and compressing the rest can preserve information while reducing bandwidth).

Early work focused on the importance of low-level image features in saliency prediction [32, 35, 36, 28, 42]. Recent approaches for saliency modeling use Convolutional Neural Networks (CNNs), Transformers, or a mixture of models [40, 41, 33, 51, 23, 25] as the backbone architecture to extract deep representations and predict the probability distribution over human gaze or fixations (computed as a Gaussian-blurred 2D heatmap, which aggregates all fixations from multiple human observers) [33, 61, 12, 42, 18]. Customized modules, such as $1\times1$ read-out convolutions [33] and Attentive ConvLSTM [18], have been introduced atop these CNNs to boost performance. Instead of the heatmap, regressing the Gaussian probability distribution has also been demonstrated as an alternative way for fixation predictions [61]. Chen et al. [12] propose to incorporate user profile information to personalize saliency predictions for each individual user.

**Scanpath prediction.** Unlike saliency, which predicts a heatmap/probability distribution over attention/importance, the goal here is to predict the sequence of eye movements as humans engage with visual content, offering insights into how individuals observe and comprehend visual information. With graphic designs as an example, predicting scanpaths can help optimize content placement, ensuring priority content captures attention first.

Prior work on human scanpath prediction has explored different task-scenarios, such as free-viewing, object searching, and visual question answering. Yang et al. [77] introduce a method utilizing inverse reinforcement learning for scanpath prediction during visual object searching. Continuing with this framework, Yang et al. [78] propose the concept of Foveated Feature Maps, enabling scanpath prediction when users search for an object that is not present in the initial image. To facilitate instruction following when performing a visual search over the image, Chen et al. [15] propose the use of a visual-question-answering model combined with a ConvLSTM module to predict the

distribution of fixation positions and duration in response to a question regarding the associated natural image. In recent work, Mondal et al. [54] propose employing the Transformer model to regress coordinates for each fixation within a scanpath dedicated to object searching. This regression is conditioned on the embedding of the object's name from a pretrained language model.

**Subjective rating prediction.** Predicting explicit human responses such as subjective preference/likes can help to better assess image quality and improve graphic designs. These responses can be continuous or discrete ratings, and may reflect both technical quality and aesthetic quality of an image. Explicit human feedback has been used in many applications. Statistic-based methods [53, 83] and Convolutional Neural Networks-based methods [84, 68] are proposed, and recently Vision Transformer has also been adopted for this task [37].

**Limitations of prior work.** While there has been significant progress in modeling behavioral tasks such as saliency, scanpaths and subjective preferences, a key limitation is that prior approaches often focused on a dedicated model for each specific task x input domain. As a result, there are saliency models for natural images, scanpath prediction models for graphic designs, or subjective ratings/likes on webpages, but there isn't a single unified model that generalizes across different tasks and domains. Instead of several dedicated per-task or per-domain models, our work seeks to build a single, unified model for these human-centered prediction tasks across diverse visual content.

**Multi-tasking unified model for language & vision.** There have been significant recent advances in large language models for natural language processing and vision-language learning [14, 57, 17, 3, 69, 62]. The underlying modeling recipe involves fine-tuning large transformer models on datasets containing a variety of recognition and reasoning tasks such as text summarization, sentiment analysis, machine translation for language models, and image captioning, question-answering, detection, and segmentation for vision-language models. These fine-tuned large models show strong generalization capacity across various tasks and data domains. Inspired by these generalizable models for language and vision, we propose UniAR – a unified model for predicting different types of human visual behavior (from attention to likes) on a variety of visual content.

## 3 Unifying Human Attention and Responses

Our model architecture along with example inputs and corresponding outputs is shown in Figure 1.

### 3.1 Model Architecture

Inspired by the recent progress in large vision-language models [14, 50, 45], We adopt a multimodal encoder-decoder transformer model to unify the various human behavior modeling tasks. The model takes two types of inputs: an image and a text prompt. Its architecture comprises of the following components: a Vision Transformer model [24] for image encoding, a word embedding layer to embed text tokens, and a T5 [60] Transformer encoder to fuse image and text representations. Additionally, it has three separate predictors: a heatmap predictor for attention/saliency heatmaps or visual importance heatmaps, a scanpath predictor for the sequence/order of viewing, and a rating predictor for quality/aesthetic scores of images or webpages. These predictors are described in Sections 3.2 to 3.4. Besides the architecture, the text prompt is designed to encode relevant information about the input domain (e.g., natural image, graphic design, webpages), the expected prediction type of the model (e.g., interaction heatmaps, sequence-of-viewing, or aesthetic score), and other task-related information such as viewing scenarios (e.g., free-viewing or object-searching), target object names, or questions to be answered, as described in Section 3.5. More details about model architecture including # of layers and layer size can be found in Appendix A.

To pretrain the model, we use both natural images from the WebLI dataset [14] and Web/mobile UI images [50], to ensure that the model can generalize to multiple domains. Image captioning and captioning for a screen region are used as the pretraining tasks, as in the original papers. To support sequence tasks involving prediction of gaze/interaction coordinates, such as scanpath prediction, we also add a pretraining task to predict the coordinates of the bounding box of relevant items given a text snippet and the screenshot (for webpage and mobile interface data).

### 3.2 Heatmap Predictor

Our model incorporates a heatmap head which is commonly used in attention/saliency research (i.e., predicting probability distribution of gaze over the input image). The heatmap prediction head takes

Table 1: List of all public datasets used to train our model. '# Image' denotes the number of unique images in the entire dataset. Note that for annotation 'scanpath,' there are multiple scanpaths recorded from a group of users associated with one image, so '# Training Sample' is much larger than '# Image.' During training, we randomly sample from all training datasets with an equal sampling rate.

| Dataset | Domain | Annotation | Viewing style | # Image | Resolution | # Training |
|---------|--------|------------|---------------|---------|-----------|-----------|
| *Salicon* [34] | Natural images | Attention heatmap | Free-viewing | 15,000 | $640 \times 480$ | 10,000 |
| *OSIE* [75] | Natural images | Attention heatmap | Free-viewing | 700 | $800 \times 600$ | 500 |
| *CAT2000* [6] | Natural images/cartoons/art... | Attention heatmap | Free-viewing | 4000 | $1920 \times 1080$ | 2000 |
| *WS-Saliency* [11] | webpage | Attention heatmap | Free-viewing | 450 | $1,280 \times 720$ | 392 |
| *Mobile UI* [47] | Mobile user interface | Attention heatmap | Free-viewing | 193 | Varied | 154 |
| *Imp1k* [27] | Graphic design | Importance heatmap | N/A | 998 | Varied | 798 |
| *WS-Scanpath* [11] | webpage | Scanpath (sequence) | Free-viewing | 450 | $1,280 \times 720$ | 5,448 |
| *FiWI* [67] | webpage | Attention heatmap | Free-viewing | 159 | $1,360 \times 768$ | 121 |
| *COCO-Search18* [16] | Natural images | Scanpath (sequence) | Object-searching | 3,101 | $1,680 \times 1,050$ | 21,622 |
| *Koniq-10k* [31] | Natural images | Subjective rating | N/A | 10,073 | $1,280 \times 720$ | 7,000 |
| *Web Aesthetics* [21] | webpage | Subjective rating | N/A | 398 | $1,280 \times 768$ | 398 |

the fused image tokens after the Transformer encoder, and processes the features via several read-out convolution layers, together with up-sampling so that the output will match the resolution of the input image. A sigmoid function is used at the end to ensure the generated values fall within the range $[0, 1]$ for each pixel.

In the experiments, we consider two different types of heatmaps, namely saliency and importance heatmap. A saliency heatmap is generated by aggregating human eye fixations from multiple participants viewing an image. On the other hand, importance heatmaps are obtained by participants highlighting or drawing bounding boxes to indicate the most critical design elements in a graphic design [27]. Each of these heatmaps reflects distinct aspects of human attention and preference.

The text prompt specifies which heatmap-type to generate for a given input sample, thereby allowing our model to predict a variety of heatmap prediction tasks (e.g., attention, interaction, importance etc.) using a single heatmap prediction head. We adopt a pixel-wise $\ell_2$ loss function for the heatmap predictor during training.

### 3.3 Scanpath (Sequence) Predictor

The scanpath predictor takes both the fused image and text tokens after the Transformer encoder as input, and applies a Transformer decoder to generate the predicted scanpath.

A scanpath is defined as a sequence of 2D locations $(x_1, y_1), (x_2, y_2), \ldots, (x_N, y_N)$ with a total of $N$ fixations, capturing the temporal aspect of attention and visual exploration. The subsequent fixations are conditional on all the previous fixations, thus fitting an autoregressive model with conditional generation. Inspired by previous literature, we use Transformer decoder for object detection and other localization tasks [13, 50], and therefore generate the location end-to-end with Transformer and text generation. In general, we let the Transformer predict the sequence of coordinates as characters in a string one after one and readout the locations from the generated text subsequently.

We spatially decompose the entire image into $1,000 \times 1,000$ bins with equal interval, and map each coordinate $x_n$ or $y_n$ to its nearest bin $\tilde{x}_n, \tilde{y}_n \in \mathbb{Z}$ in the range $[0, 999]$.

To formulate the target sequence for teacher-forcing training, we put a special token '`<extra_id_01>`' at the start of each target sequence, and attach another special token '`<extra_id_02>`' at the end, to indicate the entire scanpath sequence. We concatenate location coordinates with a separation word '`and`'. Let $y$ indicate the target sequence with length $3N + 1$ (corresponding to $N$ fixations), we have the target sequence for teacher-forcing training as follows ($\hookrightarrow$ indicates the line changing due to the paper format):

$$y = \texttt{<extra\_id\_01>} \ \tilde{x}_1 \ \tilde{y}_1 \ \texttt{and} \ \tilde{x}_2 \ \tilde{y}_2 \ \texttt{and} \ \ldots \ \texttt{and} \ \tilde{x}_N \ \tilde{y}_N \ \texttt{<extra\_id\_02>}.$$

The training objective is to maximize the log-likelihood of tokens conditioned on the input image and all preceding tokens in ground-truth scanpath string, i.e.,

$$\max \sum_{j=1}^{3N+1} w_j \log P(\tilde{y}_j | x, y_{1:j-1}), \tag{1}$$

where $x$ is the input image and text prompt, and $y$ is the target sequence associated with $x$. $w_j$ is the weight for the $j$-th token. We use a unified weight for each token in experiments.

**Decoding during inference.** During inference, it is not strictly guaranteed that the generated string will exactly follow the format of the target sequence, especially when the number of fixations is relatively large (e.g., $N \geq 30$), resulting in an invalid sequence to readout. To deal with the possible invalid cases, in practice, we identify two special tokens '`<extra_id_01>`' and '`<extra_id_02>`' in the predicted string if available, and extract the context between these two special tokens. Then we split the extracted string with the separator word '`and`'. For a pair of two tokens from the beginning, we check if they are both numerical. If so, we add one fixation with this coordinate after mapping them to the original resolution, then iteratively move to the following, and if not, we terminate the decoding process and keep the existing sequence. If there is no fixation available in the predicted string, we mark the scanpath as invalid. During training, we observe that the valid rate (#valid scanpaths / #scanpaths generated) of scanpath decoding quickly converges to 1, meaning every predicted scanpath will contain valid fixation(s).

Compared to the scanpath predictors in GazeFormer [54] which predicts each point as 2D continuous numbers instead of two text tokens, our model seems less intuitive. However, as shown in Section 4.3, the proposed scanpath predictor works quite well. Moreover, one advantage of the current design of the scanpath predictor is that it can be easily extended to predict other types of human behavior sequences, e.g., text sequence, or 1-D number sequence, with minor modifications on the sequence output format. This flexibility is important for a unified model like ours.

### 3.4 Rating Predictor

This prediction head takes image tokens after the Transformer encoder module, and processes the features via a few convolution and connected layers. An $\ell_2$ loss is used for training the rating predictor with rating data.

### 3.5 Text Prompt

To enhance the model's ability to generalize across a variety of visual content and task scenarios, we integrate specific task instructions into the model via text prompts. The prompts used in UniAR are structured as follows:

INPUT_TYPE:<input_type> OUTPUT_TYPE:<output_type> QUERY:<query>.

We fill `<input_type>` with string taken from {`natural image` | `webpage` | `graphic design` | `mobile user interface`} and `<output_type>` taken from {`saliency heatmap` | `importance heatmap` | `aesthetics score` | `scanpath`}. We append a query in string `Query:<query>` to the prompt if a task-specific query is available, for example, the object name to search, or the question to answer, depending on the use case. For example, an example full prompt is "INPUT_TYPE: natural image OUTPUT_TYPE: scanpath QUERY:searching a bowl", which guides the model to predict scanpath output on a natural image under the task of "searching a bowl". The prompt we use is modularized and can easily adapt to different types of datasets and scenarios.

## 4 Experiment

### 4.1 Protocol

**Datasets.** Please refer to Table 1 for all public datasets we consider in training and benchmarking. For more dataset processing details, please refer to Appendix B.

**Benchmarks.** We reuse benchmarks from recent literature for model comparison purposes. We adopt the benchmarks for *WS-Saliency* and *WS-Scanpath* from Tables 3 and 7 in Chakraborty et al. [11] respectively, *Mobile UI* from Table 2 in Leiva et al. [47], *Imp1k* from Table 2 in Fosco et al. [27], *OSIE* from Table 4 in Chen et al. [12], *Salicon* from Table 1 [4] in Reddy et al. [61], *COCO-Search18* from Table 1 in Mondal et al. [54], *KonIQ-10k* from Table 2 in Ke et al. [37], and Web Aesthetics from Table 4 in Delitzas et al. [21]. We also provide model results from some other papers for comparison [25, 33]. The baseline results for *CAT2000* are from *CAT2000* Leaderboard [5].

---

[4]There are two versions of Salicon data: Salicon 2015 and Salicon 2017. The results are on Salicon 2017.

[5]CAT2000 Leaderboard: https://saliency.tuebingen.ai/results_CAT2000.html

**Evaluation metrics.** Inheriting from the above benchmarks, we consider the following evaluation metrics. **CC** [44]: Pearson's Correlation Coefficient is used here to measure the linear relationship in all pixel values between the predicted and ground-truth saliency heatmaps; **KLD** [39]: the metric to use KL-Divergence between the predicted heatmap and ground-truth heatmap to measure the distribution discrepancy, with the prediction used as the target distribution. **AUC-Judd** [35]: Area under ROC curve (AUC) in the variant from Judd et al. [35] treating the heatmap prediction as binary classification with various thresholds. The specific calculations of true positive and false positive rates can be referred to [10]. **sAUC** [7]: the shuffled AUC metric samples negatives from other images for AUC calculation. **NSS** [59]: Normalized Scanpath Saliency is the average saliency strength (pixel values in the predicted heatmap) at all ground-truth fixation locations. **SIM** [64]: Similarity is computed as the sum of the minimum values among the normalized prediction and ground-truth heatmaps. **RMSE**: the root mean square error between the predicted and ground-truth heatmaps. **R-Squared** ($R^2$): the coefficient of determination applied to all values in the heatmap. **SemSS** [78]: Semantic Sequence Score converts each fixation to an ID decided by a semantic segmentation over the image, and compares two strings with a string-matching algorithm [56]. **SemFED** [78]: similar to SemSS, Semantic Fixation Edit Distance uses Levenshtein distance for string matching [48]. **SequenceScore**: similar to SemSS, but instead of using a semantic segmentation map, the Sequence Score uses the clustering results from a MeanShift clustering to segment the image and map the fixation to their ID. **MultiMatch** [22]: MultiMatch is the average of four metrics of scanpath, namely: **Shape**, **Direction**, **Length**, and **Position**, characterizing the similarity between the predicted scanpath and its ground-truth. **SRCC** and **PLCC**: refer to Spearman's rank correlation coefficient and Pearson's linear correlation coefficient, respectively, used to quantify the quality of predicted ratings.

Experimental benchmarks for one task among different datasets may not have uniform evaluation metrics but most of their metrics are shared. For saliency and importance heatmap predictions, we resize the predicted heatmap back to its original image resolution for evaluation.

## 4.2 Model Training

We pretrain the model on a series of pre-training tasks, including Web/Mobile UI understanding [50] and natural image captioning [14]. Subsequently, we fine-tune the entire model using the Adafactor optimizer with a learning rate of 0.1, batch size of 128, and image resolution of 512×512. All images maintain their aspect ratio and are padded to fit the training resolution. The model uses ViT B16 as the vision encoder and T5 base as the Transformer encoder of the image and text tokens, resulting in a total of 848 million parameters. The model is implemented in JAX. We use 64 Google Cloud TPU v3 to train UniAR for 20k steps in 12 hours.

**Datasets mixture.** As we are combining a series of public datasets, in every iteration, for all training datasets in Table 1, we employ a random sampling strategy that ensures an equal sampling rate across these datasets. This approach guarantees that each dataset has an equal probability of contributing a sample, irrespective of its sample volume.

Table 2: Subjective rating prediction results on Natural image image dataset *KonIQ-10k* and webpage dataset *Web Aesthetics*.

| Dataset | Method | SRCC ↑ | PLCC ↑ |
|---|---|---|---|
| *KonIQ-10k* [31] (Natural image) | BRISQUE [53]'12 | 0.665 | 0.681 |
| | ILNIQE [83]'15 | 0.507 | 0.523 |
| | HOSA [74]'16 | 0.671 | 0.694 |
| | BIECON [38]'16 | 0.618 | 0.651 |
| | WaDIQaM [8]'17 | 0.797 | 0.805 |
| | PQR [81]'17 | 0.880 | 0.884 |
| | SFA [49]'18 | 0.856 | 0.872 |
| | DBCNN [84]'18 | 0.875 | 0.884 |
| | MetaIQA [86]'20 | 0.850 | 0.887 |
| | BIQA (25 crops) [68]'20 | **0.906** | 0.917 |
| | MUSIQ-single [37]'21 | 0.905 | 0.919 |
| | UniAR | 0.905 -0.11% | **0.925** +0.65% |
| *Web Aesthetics* [21] (webpage) | Rating-based Calista [21]'23 | - | 0.770 |
| | Comparison-based Calista [21]'23 | - | 0.820 |
| | UniAR | **0.811** | **0.839** +2.31% |

## 4.3 Experiment Results

We present the results of UniAR for predicting heatmaps, scanpath-sequences as well as ratings across domains and datasets in Tables 2 to 4, in comparison with the baselines that are trained on a specific domain, task, or dataset.

**Heatmap prediction.** Table 3 shows the performance of UniAR across 7 public benchmarks. A complete version of results including all baselines and metrics is presented in Table 6 in Appendix C. Among these datasets, which vary in domains (natural images, webpages, and graphic designs) and tasks (heatmaps of attention, and importance), UniAR achieves SOTA performance compared to strong baselines, and outperforms previous SOTAs in many cases. On *Mobile UI* and *Imp1k* datasets, UniAR outperforms previous SOTA across every metric. Out of the 27 metrics listed in Table 3,

Table 3: Heatmap prediction results on 7 public datasets across natural images, art, cartoons, mobile UIs, and webpages (Please refer to Table 6 in Appendix C for complete baselines & metrics). For *Imp1k* we predict the importance heatmap, while for the remaining datasets, we predict the attention/saliency heatmap. For each dataset and metric, the best result is in **bold**, second best is in blue, and our method is highlighted in green. For our model, the relative performance change compared to the best result is noted. Note that the metric values for baseline models are obtained from existing references as described in the "Benchmarks" paragraph. "-" means the metrics are not reported in references. Also note that there are two versions of Salicon data, Salicon 2015 and Salicon 2017. The results in this table are on Salicon 2017.

| Dataset | Method | CC ↑ | KLD ↓ | AUC-Judd ↑ | NSS ↑ |
|---|---|---|---|---|---|
| *WS-Saliency* [11] (Webpage) | SAM-ResNet [18]'18 | 0.596 | 1.506 | 0.795 | 1.284 |
| | EML-NET [33]'20 | 0.565 | 2.110 | 0.790 | 1.277 |
| | UMSI [27]'20 | 0.444 | 1.335 | 0.757 | 1.042 |
| | DI Net + *WS* [76]'19 | 0.798 | 0.690 | 0.852 | 1.777 |
| | AGD-F (W/o-L) [11]'22 | 0.815 | 0.637 | 0.858 | **1.802** |
| | UniAR | **0.827** +1.47% | **0.299** -53.06% | **0.860** +0.23% | 1.783 -1.05% |
| *FiWI* [67] (Webpage) | AGD-F [11]'22 | **0.735** | - | 0.767 | 1.606 |
| | EML-NET [33]'20 | 0.661 | 0.603 | 0.847 | 1.653 |
| | EML-NET + *Salicon* [33]'20 | 0.689 | 0.567 | 0.848 | 1.722 |
| | Chen et al. [12]'23 | 0.699 | 0.564 | 0.851 | 1.752 |
| | UniAR | 0.734 -0.14% | 0.571 +0.29% | **0.859** +0.94% | **1.838** +4.91% |
| *Mobile UI* [47] (Mobile interface) | ResNet-Sal [47]'20 | 0.657 | - | 0.692 | 0.704 |
| | SAM-S2015 [18]'18 | 0.477 | - | 0.650 | 0.537 |
| | SAM-S2017 [18]'18 | 0.834 | - | 0.723 | 0.839 |
| | SAM-mobile [47]'20 | 0.621 | - | 0.666 | 0.655 |
| | UniAR | **0.879** +5.40% | **0.115** | **0.756** +4.56% | **1.008** +20.14% |
| *CAT2000* [43] (Natural images, cartoons, art...) | DeepGaze II [42]'17 | 0.795 | 0.382 | 0.864 | 1.962 |
| | UNISAL [25]'20 | 0.740 | 0.470 | 0.860 | 1.936 |
| | DeepGaze IIE [51]'21 | 0.819 | **0.345** | 0.869 | 2.112 |
| | SalFBNet [23]'22 | 0.703 | 1.198 | 0.855 | 1.879 |
| | UniAR | **0.870** +6.23% | 0.613 +77.68% | **0.877** +0.92% | **2.338** +10.70% |
| *Salicon* [34] (Natural images) | SimpleNet w. ResNet-50 [61]'20 | 0.895 | 0.211 | 0.868 | 1.881 |
| | SimpleNet w. PNASNet-5 [61]'20 | **0.907** | **0.193** | **0.871** | 1.926 |
| | MDNSal [61]'20 | 0.899 | 0.217 | 0.868 | 1.893 |
| | UNISAL [25]'20 | 0.880 | 0.226 | 0.867 | 1.923 |
| | EML-NET [33]'20 | 0.890 | 0.204 | 0.802 | **2.024** |
| | UniAR | 0.900 -0.77% | 0.214 +10.88% | 0.870 -0.11% | 1.946 -3.85% |
| *OSIE* [70] (Natural images) | SAM-ResNet [18]'18 | 0.758 | **0.480** | 0.860 | 1.811 |
| | UMSI [27]'20 | 0.746 | 0.513 | 0.856 | 1.788 |
| | EML-NET [33]'20 | 0.717 | 0.537 | 0.854 | 1.737 |
| | Chen et al. [12]'23 | **0.761** | 0.506 | 0.860 | **1.840** |
| | UniAR | 0.742 -2.50% | 0.583 +21.46% | **0.862** +0.23% | 1.789 -2.77% |

| Dataset | Method | CC ↑ | KLD ↓ | RMSE ↓ | $R^2$ ↑ |
|---|---|---|---|---|---|
| *Imp1k* [27] (Graphic design) | SAM [18]'18 | 0.866 | 0.166 | 0.168 | 0.108 |
| | UMSI-nc [27]'20 | 0.802 | 0.177 | 0.152 | 0.095 |
| | UMSI-2stream [27]'20 | 0.852 | 0.168 | 0.141 | 0.105 |
| | UMSI [27]'20 | 0.875 | 0.164 | 0.134 | 0.115 |
| | UniAR | **0.904** +3.31% | **0.124** -25.00% | **0.079** -41.04% | **0.823** +615.65% |

UniAR achieves the best result in 17 of them and the second best in 6 cases. In summary, UniAR experimentally demonstrates promising performance in saliency modeling in various fields.

**Scanpath prediction.** In Table 4, scanpath-sequence prediction results are shown for two datasets: COCO-Search18 [16] (scanpath in natural images for object searching) and *WS-Scanpath* [11] (scanpath on webpages under free viewing). On both the datasets, UniAR performs comparably to baselines, and further outperforms the baselines on all the metrics on *WS-Scanpath*. Among the 5 reported metrics in Table 4, our model achieved the best result in 4 of them. A complete version of results including all baselines and metrics is presented in Table 7 in Appendix C.

**Score prediction.** In Table 2, we present rating prediction results on two datasets: *KonIQ-10k* [31] on natural images and *Web Aesthetics* [21] on webpages. UniAR achieves the best results for PLCC metrics on both datasets and the second best for SRCC on *KonIQ-10k*. Note that in Ke et al. [37], a multi-scale version of MUSIQ performs slightly better than UniAR on SRCC (0.916 vs 0.905). However, since UniAR does not use multi-scale inputs, we did not include those results.

Table 4: Scanpath (sequence) prediction results on natural image and digital design datasets. Please refer to Table 7 in Appendix C for complete baselines & metrics.

| Dataset | Method | SemSS ↑ | SemFED ↓ | Sequence Score ↑ | MultiMatch ↑ |
|---|---|---|---|---|---|
| *COCO-Search18* [16] (Natural images, object searching) | IRL [77]'20 | 0.481 | 2.259 | - | 0.833 |
| | Chen et al. [15]'21 | 0.470 | 1.898 | - | 0.820 |
| | FFM [78]'22 | 0.407 | 2.425 | - | 0.808 |
| | Gazeformer [54]'23 | 0.496 | 1.861 | - | 0.849 |
| | UniAR | **0.521** +5.04% | 2.004 +7.68% | - | **0.874** +2.94% |
| *WS Scanpath* [11] (webpage, free-viewing) | SceneWalker [66]'20 | - | - | 0.194 | 0.716 |
| | AGD-F (w. layout) [11]'22 | - | - | 0.203 | 0.719 |
| | AGD-S (w/o layout) [11]'22 | - | - | 0.221 | 0.745 |
| | AGD-S (w. layout) [11]'22 | - | - | 0.224 | 0.755 |
| | UniAR | - | - | **0.267** +19.20% | **0.887** +17.48% |

**Transferring knowledge between tasks.** We test UniAR's ability to generalize and transfer to unseen tasks/domain combinations. Our model is trained on certain combinations of task and image domains, and tested on new, unseen combinations of behavior tasks and image domains.

Table 5: Experiments on transferring knowledge from other domain/task combinations to *WS-Scanpath* dataset for scanpath predictions. *CC = COCO-FreeView* dataset.

| Training Set | Sequence Score ↑ | MultiMatch ↑ |
|---|---|---|
| *WS-Scanpath* (previous SoTA [11]) | 0.224 | 0.755 |
| *WS-Scanpath* (ours) | 0.261 | 0.894 |
| UniAR full model | 0.267 | 0.887 |
| *CC* scanpath (ours) | 0.196 | 0.836 |
| *CC* scanpath + *WS-Saliency* (ours) | 0.190 | 0.858 |
| *CC* saliency/scanpath + *WS-Saliency* (ours) | 0.231 | 0.857 |

Our experiment uses *WS* (webpage) and *COCO-Freeview* (free-viewing on natural image) datasets, where WS-saliency, WS-scanpath, CC saliency and CC scanpath, are saliency and scanpath data for WS and COCO-Freeview data respectively. We test the model performance on scanpath prediction on the webpage scanpath data (*WS-Scanpath* dataset). We consider three different training scenarios: (1) Using scanpath data from natural image (*COCO-Freeview*); (2) Combining scanpath from natural image (*COCO-Freeview*) with saliency heatmaps from webpage (*WS*); (3) Employing both scanpath and saliency heatmap from natural image (*COCO-Freeview*), augmented with saliency heatmap from webpage (*WS*). Each scenario maintains some relevance to our test set by either sharing the same task or image domain, but never both.

In Table 5, we show experimental results for the above scenarios, and also attach the baseline results on this test set and the results from UniAR (full training data) as reference. As shown in Table 5, our third training scenario: leveraging scanpath and saliency heatmaps from *COCO-Freeview* and saliency heatmap from *WS*, shows good results against previous SoTA [11], despite the model not having seen webpage scanpath data during training. Prediction performance declines in scenarios 1 & 2, which take more limited datasets, but remain competitive.

## 5 Limitations and Future Work

When modeling human preferences and behavior, it is important to carefully consider ethics and AI principles, and acknowledge dataset limitations.

**Ethics and AI principles.** Modeling any aspect of human behavior should adhere to ethical guidelines on data collection and applications, and be conducted in a transparent way, including clarifying the limitations of the model when replicating human preferences. It should keep humans in the loop, as the model prediction is intended as a reference, not as a means to replace evolving human preferences with a synthetic guide. We take these ethical considerations and AI Principles into account, ensuring that the model usage remains socially beneficial and responsible.

**Aligning with human preference.** While UniAR is trained to predict human preferences and behavior, we recognize that using it as a reward model may lead to reward hacking, which would make it less representative of genuine human preference. We suggest considering techniques [46, 26] to mitigate this.

**Representing diverse human preferences.** Humans have diverse preferences for subjective notions like image attractiveness. Without personalization, the model converges towards a more uniform notion of preference - a common concern for ML models. To promote visual diversity when using UniAR, we propose two strategies: (1) using the model in a hybrid manner, providing insights to

help humans make decisions in applications like web or visual content optimization, and (2) develop personalized models based on our initial unified model, which will help generate more diverse predictions based on user attributes.

**Adjusting to evolving human preferences.** Human preferences evolve over time, and to remain accurate, the model has to adjust accordingly. Updates can be achieved by fine-tuning with more recent data. Future work will include updating the training data, and exploring concepts like continual learning techniques [72], to keep the model up to date.

**Dataset limitations.** Using diverse, representative datasets is important to minimize potential biases in the model. In this paper, we focus on a proof-of-concept for unified modeling of human attention/preference behavior, based on existing, publicly available datasets. Below is a listing of annotator demographics, as described in the original papers.

1. *WS-saliency* [11]: "A total of 41 participants (19 females, 22 males; age range 17-23; with normal or corrected-to-normal vision) participated in our data collection."

2. *Mobile UI* [47]: "Thirty participants (12 male, 18 female). [...] The average age was 25.9 (SD=3.95). The participants had normal vision (8) or corrected-to-normal-vision (22). Twenty of the 22 wore glasses and the remaining two wore contact lenses."

3. *Imp1k* [27]: "The data of 43 participants (29 male, most in their 20s and 30s) were used in the resulting analyses."

4. *FiWI* [67]: "11 students (4 males and 7 females) in the age range of 21 to 25 participated in data collection. All participants had normal vision or corrective visual apparatus."

Despite some balance in male vs. female participants, the age distribution is skewed towards participants in their 20s. This is likely because most data were collected at universities. Crowdsourced datasets like Salicon and Koniq-10K are expected to cover a wider range of age and other attributes. Future work will focus on gathering a more diverse and representative dataset.

**Improve accessibility.** UniAR predicts human attention, trained on datasets collected from humans not experiencing visual impairments beyond corrective lenses. UniAR cannot model behavior of, for example, blind and low-vision users directly, but it can still benefit them by acting as an accessibility tool for highlighting important areas of a webpage for a screen reader. One way to enhance the accessibility of the model is using multi-modal preference modeling, which incorporates not just visual cues, but also how users interact with content through screen readers, voice commands, and other assistive technologies. Collaborating with accessibility experts and organizations could help improve future iterations of our work.

## 6 Conclusion

We developed a multimodal, unified model UniAR to predict different types of implicit and explicit human behavior on visual content, from attention to subjective preferences/likes, using image and text prompts. This model, trained on diverse public datasets across natural images, graphic designs, webpages and UIs, effectively predicted human attention heatmaps, scanpath sequences, and aesthetic or quality scores. Our model achieved SOTA performance across multiple benchmarks and tasks. We plan to explore more behavior tasks and domains in future work.

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

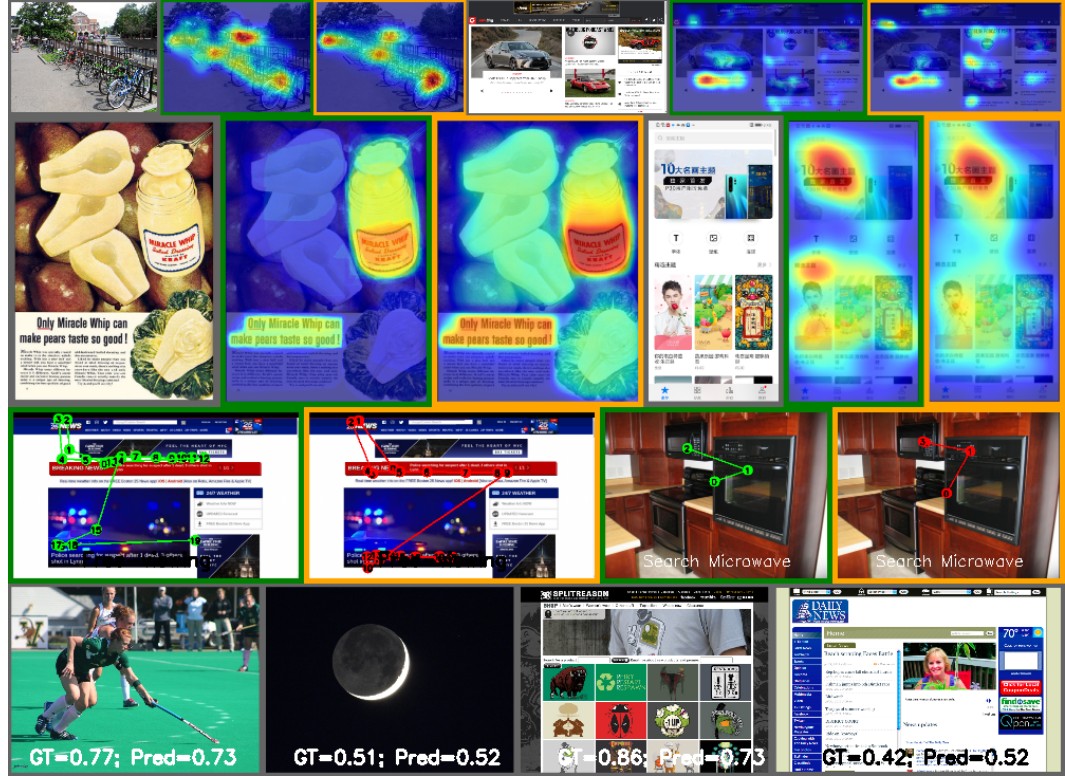

Figure 3: Another set of visualizations on UniAR's predictions. Images in green border are ground-truth, while images in orange border are UniAR's predictions. **First row**: saliency heatmap on *Salicon* and *WS-Saliency*. **Second row**: importance heatmap on *Imp1k*, and saliency heatmap on *Mobile UI*. **Third row**: free-viewing scanpath on *WS-Scanpath* and object-searching scanpath on *COCO-Search18*. **Fourth row**: rating prediction on *Koniq-10k* and *Web Aesthetics* datasets.

## A   Model Details

The main model components consist of a ViT B16 encoder for image encoding, a T5 base encoder for mixing image and text tokens, and three predictors for rating, heatmap, and scanpath prediction, respectively.

**Vision Transformer and T5 Encoder.**   The ViT B16 encoder uses $16 \times 16$ patch size, 12 layers with 12 heads, MLP dimension 3,072, and hidden dimension 768. The T5 base encoder uses 12 layers with 12 heads and MLP dimension 2,048 and hidden dimension 768.

**Score Predictor.**   The score predictor consists of four convolutional layers with Layer Normalization and ReLU activation. The filter size, kernel size, and strides are $[768, 384, 128, 64]$, $[2, 2, 2, 2]$, $[1, 1, 1, 1]$, respectively. Three dense layers of size 2,048, 1,024 and 1 are used to generate a scalar with ReLU activations for the first two layers, and sigmoid for the last.

**Heatmap Predictor.**   The heatmap predictor consists of two convolution layers with filter size, kernel size, and stride as $[768, 384]$, $[3, 3]$, $[1, 1]$, respectively. It then uses four de-convolution layers to up-sample to the required output resolution, with the filter size, kernel size, and stride as $[768, 384, 384, 192]$, $[3, 3, 3, 3]$, $[2, 2, 2, 2]$, respectively. Each de-convolution layer is with two read-out convolution layers of kernel size 3 and stride 1. Layer Normalization and ReLU are used for each layer. In the end, two read-out convolution layers and a final sigmoid activation are used to generate the heatmap prediction.

**Scanpath Predictor.**   The scanpath predictor is implemented using a T5 base decoder with 12 layers of 12 heads and MLP dimension 2,048 and hidden dim 768. Output token length is 64.

We combine the losses from the three predictors, i.e., sequence cross-entropy loss, heatmap L2 loss, and score L2 loss using weights [1, 500, 50] empirically (to make them at the similar scale).

## B   Dataset Processing

In this section, we describe some of the key dataset processing details.

Table 6: The full table for heatmap prediction results on 7 public datasets spanning digital design and natural images, benchmarking with eight metrics in total. The details on datasets and evaluation metrics can be found in Section 4.1. For *Imp1k* dataset we predict the importance heatmap, while for the rest of the four datasets, we predict attention/saliency heatmap. Our method is highlighted with green background . For each dataset and each metric, the best result in the current column is in **bold**, and the second best result is in blue. For our model, the relative performance change compared to the second best result (or the best result if we are not the best) in % is noted.

| Dataset | Method | CC ↑ | KLD ↓ | AUC-Judd ↑ | sAUC ↑ | SIM ↑ | NSS ↑ | RMSE ↓ | $R^2$ ↑ |
|---|---|---|---|---|---|---|---|---|---|
| *Mobile UI* [47] (Mobile interface) | Itti et al. [32]'98 | 0.082 | - | 0.223 | - | 0.558 | 0.126 | - | - |
| | BMS [82]'13 | 0.131 | - | 0.249 | - | 0.206 | 0.138 | - | - |
| | GBVS [28]'06 | 0.580 | - | 0.666 | - | 0.709 | 0.591 | - | - |
| | ResNet-Sal [47]'20 | 0.657 | - | 0.692 | - | 0.734 | 0.704 | - | - |
| | SAM-S2015 [18]'18 | 0.477 | - | 0.650 | - | 0.562 | 0.537 | - | - |
| | SAM-S2017 [18]'18 | 0.834 | - | 0.723 | - | 0.819 | 0.839 | - | - |
| | SAM-mobile [47]'20 | 0.621 | - | 0.666 | - | 0.664 | 0.655 | - | - |
| | UniAR | **0.879** +5.40% | **0.115** | **0.756** +4.56% | - | **0.832** +1.59% | **1.008** +20.14% | 0.116 | 0.777 |
| *WS-Saliency* [11] (webpage) | Itti et al. [32]'98 | 0.367 | 0.840 | 0.710 | 0.661 | - | 0.769 | | |
| | Deep Gaze II [41]'16 | 0.574 | 3.449 | 0.815 | 0.644 | - | 1.380 | - | - |
| | SalGAN + *WS* [58]'17 | 0.637 | 0.622 | 0.818 | 0.703 | - | 1.458 | - | - |
| | DVA [73]'17 | 0.571 | 0.701 | 0.805 | 0.711 | - | 1.260 | - | - |
| | UAVDVSM [29]'19 | 0.519 | 0.858 | 0.739 | 0.668 | - | 1.133 | - | - |
| | SAM-ResNet [18]'18 | 0.596 | 1.506 | 0.795 | 0.717 | - | 1.284 | - | - |
| | EML-NET [33]'20 | 0.565 | 2.110 | 0.790 | 0.702 | - | 1.277 | - | - |
| | UMSI [27]'20 | 0.444 | 1.335 | 0.757 | 0.698 | - | 1.042 | - | - |
| | TaskWebSal-FreeView [85]'18 | 0.525 | 0.784 | 0.769 | 0.714 | - | 1.107 | - | - |
| | SAM-ResNet + *WS* [18]'18 | 0.718 | 0.994 | 0.828 | 0.725 | - | 1.532 | - | - |
| | DI Net + *WS* [76]'19 | 0.798 | 0.690 | 0.852 | 0.739 | - | 1.777 | - | - |
| | AGD-F (W/o-L) [11]'22 | 0.815 | 0.637 | 0.858 | 0.753 | - | **1.802** | - | - |
| | UniAR | 0.827 +1.47% | 0.299 -53.06% | 0.860 +0.23% | 0.779 +3.45% | 0.737 | 1.783 -1.05% | 0.085 | 0.691 |
| *FiWI* [67] (webpage) | DeepGaze II [41]'16 | 0.488 | - | 0.797 | 0.625 | - | 1.229 | - | - |
| | SAM-ResNet [18]'18 | 0.595 | - | 0.791 | 0.673 | - | 1.246 | - | - |
| | UMSI [27]'20 | 0.457 | - | 0.755 | 0.675 | - | 0.938 | - | - |
| | AGD-F [11]'22 | **0.735** | - | 0.767 | 0.748 | - | 1.606 | - | - |
| | EML-NET [33]'20 | 0.661 | 0.603 | 0.847 | 0.675 | - | 1.653 | - | - |
| | EML-NET + *Salicon* [33]'20 | 0.689 | 0.567 | 0.848 | 0.697 | - | 1.722 | - | - |
| | Chen et al. [12]'23 | 0.699 | 0.564 | 0.851 | 0.704 | - | 1.752 | - | - |
| | UniAR | 0.734 -0.14% | 0.571 +0.29% | 0.859 +0.94% | 0.749 +0.13% | 0.627 | 1.838 +4.91% | 0.082 | 0.544 |
| *Salicon* [34] (Natural images) | SimpleNet w. ResNet-50 [61]'20 | 0.895 | 0.211 | 0.868 | - | 0.786 | 1.881 | - | - |
| | SimpleNet w. PNASNet-5 [61]'20 | **0.907** | **0.193** | **0.871** | - | **0.797** | 1.926 | - | - |
| | MDNSal [61]'20 | 0.899 | 0.217 | 0.868 | - | 0.797 | 1.893 | - | - |
| | UNISAL [25]'20 | 0.880 | 0.226 | 0.867 | 0.725 | 0.771 | 1.923 | - | - |
| | EML-NET [25]'20 | 0.890 | 0.204 | 0.802 | 0.778 | 0.785 | 2.024 | - | - |
| | UniAR | 0.900 -0.77% | 0.214 +10.88% | 0.870 -0.11% | 0.753 -3.32% | 0.791 -0.75% | 1.946 -3.85% | 0.079 | 0.823 |
| *OSIE* [70] (Natural images) | SALICON [34]'15 | 0.685 | 0.575 | 0.846 | - | 0.600 | 1.641 | - | - |
| | SAM-ResNet [18]'18 | 0.758 | **0.480** | 0.860 | - | 0.648 | 1.811 | - | - |
| | UMSI [27]'20 | 0.746 | 0.513 | 0.856 | - | 0.631 | 1.788 | - | - |
| | EML-NET [33]'20 | 0.717 | 0.537 | 0.854 | - | 0.619 | 1.737 | - | - |
| | Chen et al. [12]'23 | **0.761** | 0.506 | 0.860 | - | **0.652** | **1.840** | - | - |
| | UniAR | 0.742 -2.50% | 0.583 +21.46% | 0.862 +0.23% | 0.745 | 0.640 -1.84% | 1.789 -2.77% | 0.103 | 0.558 |
| *CAT2000* [43] (Natural image) | ICF [42]'17 | 0.780 | 0.445 | 0.856 | 0.619 | 0.670 | 1.959 | - | - |
| | DeepGaze II [42]'17 | 0.795 | 0.382 | 0.864 | 0.650 | 0.687 | 1.962 | - | - |
| | UNISAL [25]'20 | 0.740 | 0.470 | 0.860 | 0.668 | 0.663 | 1.936 | - | - |
| | DeepGaze IIE [51]'21 | 0.819 | **0.345** | 0.869 | 0.668 | 0.706 | 2.112 | - | - |
| | SalFBNet [23]'22 | 0.703 | 1.198 | 0.855 | 0.633 | 0.643 | 1.879 | - | - |
| | UniAR | 0.870 -0.92% | 0.613 +13.96% | 0.877 +0.81% | 0.615 -7.93% | 0.752 +6.52% | 2.338 +10.70% | - | - |
| | Gold Standard | 0.969 | 0.089 | 0.916 | 0.787 | 0.866 | 2.743 | - | - |
| *Imp1k* [27] (Graphic design) | Bylinskii et al. [9]'17 | 0.758 | 0.301 | - | - | - | - | 0.181 | 0.072 |
| | Bylinskii et al. [9]'17 | 0.732 | 0.388 | - | - | - | - | 0.205 | 0.061 |
| | SAM [18]'18 | 0.866 | 0.166 | - | - | - | - | 0.168 | 0.108 |
| | UMSI-nc [27]'20 | 0.802 | 0.177 | - | - | - | - | 0.152 | 0.095 |
| | UMSI-2stream [27]'20 | 0.852 | 0.168 | - | - | - | - | 0.141 | 0.105 |
| | UMSI [27]'20 | 0.875 | 0.164 | - | - | - | - | 0.134 | 0.115 |
| | UniAR | **0.904** +3.31% | **0.124** -25.00% | - | - | 0.836 | - | 0.079 -41.04% | 0.823 +615.65% |

***Imp1k.*** In the *Imp1k* dataset, we observe some resolution mismatch between the image and its ground-truth importance map. To unify the image and the ground-truth into the same resolution, we find out the lower resolution (with a smaller area) between these two and downsample the larger one into the lower resolution.

***WS-Scanpath.*** We asked Chakraborty et al. [11] for their code on evaluating the scanpath, and follow them to MeanShift clustering to generate spatial bins for the metric **SequenceScore**. We also follow their criteria to select the number of clusters in MeanShift clustering which will be used in the evaluation later on.

***Mobile UI.*** We contacted the authors but the original dataset partition is missing. We randomly partitioned the dataset into a training and a testing set with the number of instances following Leiva et al. [47]. We calculate the saliency results using images without padding.

***COCO-Search18.*** For the scanpath results, we follow the updated experimental results in the appendix of GazeFormer [54]. The original results in the main paper are impacted by an image padding issue, as reported in their GitHub repo.

## C   Full Results and More Visualizations

We attach another set of visualizations of UniAR's predictions in Figure 3 including heatmap, scanpath, and rating predictions.

Table 7: The full table for scanpath prediction results on natural image and digital design datasets, with object-searching and free-viewing tasks.

| Dataset | Method | SemSS ↑ | SemFED ↓ | Sequence Score ↑ | Shape ↑ | Direction ↑ | Length ↑ | Position ↑ | MultiMatch ↑ |
|---|---|---|---|---|---|---|---|---|---|
| *COCO-Search18* [16] (Natural image, object searching) | IRL [77]'20 | 0.481 | 2.259 | - | 0.901 | 0.642 | 0.888 | 0.802 | 0.833 |
| | Chen et al. [15]'21 | 0.470 | 1.898 | - | 0.903 | 0.591 | 0.891 | 0.865 | 0.820 |
| | FFM [78]'22 | 0.407 | 2.425 | - | 0.896 | 0.615 | 0.893 | 0.850 | 0.808 |
| | Gazeformer [54]'23 | 0.496 | 1.861 | - | 0.905 | 0.721 | 0.857 | **0.914** | 0.849 |
| | UniAR | **0.521** +5.04% | 2.004 +7.68% | - | **0.946** +4.53% | **0.724** +0.42% | **0.924** +3.47% | 0.901 -1.42% | **0.874** +2.94% |
| *WS Scanpath* [11] (webpage, free-viewing) | Itti et al. [32]'98 | - | - | 0.177 | 0.781 | 0.676 | 0.778 | 0.594 | 0.707 |
| | MASC [2]'17 | - | - | 0.169 | 0.788 | 0.580 | 0.818 | 0.514 | 0.717 |
| | SceneWalker [66]'20 | - | - | 0.194 | 0.843 | 0.616 | 0.842 | 0.562 | 0.716 |
| | G-Eymol [80]'19 | - | - | 0.218 | 0.820 | 0.673 | 0.816 | 0.681 | 0.748 |
| | AGD-F (w. layout) [11]'22 | - | - | 0.203 | 0.787 | 0.642 | 0.771 | 0.677 | 0.719 |
| | AGD-S (w/o layout) [11]'22 | - | - | 0.221 | 0.814 | 0.663 | 0.805 | 0.698 | 0.745 |
| | AGD-S (w. layout) [11]'22 | - | - | 0.224 | 0.820 | 0.677 | 0.813 | 0.708 | 0.755 |
| | UniAR | - | - | **0.267** +19.20% | **0.967** +14.71% | **0.826** +22.01% | **0.960** +14.01% | **0.794** +12.15% | **0.887** +17.48% |

We present full tables of UniAR's performance on heatmap and scanpath predictions in Tables 6 and 7, with more baselines and a complete set of evaluation metrics. UniAR offers consistently good predictions on three tasks across multiple datasets, compared to the ground-truths.

