# OpenReview forum: "UniAR: A Unified model for predicting human Attention and Responses on visual content"
_NeurIPS.cc/2024/Conference — NeurIPS 2024 poster_

### Official Review · Reviewer_Fbj6 · 2024-07-10

**Soundness:** 2
**Presentation:** 3
**Contribution:** 2
**Rating:** 3
**Confidence:** 4

**Summary:**

In this paper, the authors introduce a novel text-image framework designed to integrate various human-response tasks and multiple image domains. These tasks include attention map generation, scanpath prediction, and subjective preference evaluation, applied to images such as webpages, natural scenes, graphic designs, and mobile user interfaces. The framework utilizes a transformer architecture, accepting both text and image inputs. Text inputs specify the tasks, while the model itself comprises a transformer encoder and three predictors (heatmap, rating, and scanpath) to generate 3 types of outputs. Initially pretrained on foundational text-image tasks, the model is fine-tuned for diverse human-response tasks

**Strengths:**

1.	The question of unifying human response-related vision models is novel and holds significant potential, particularly if the advantages of integrating tasks are thoroughly explored.
2.	The paper is well-organized and easy to understand.

**Weaknesses:**

1.	I have many concerns on the experimental results of this paper, where the authors try to demonstrate the superiority of their model performance by comparing with other methods. My concerns are as follows.

a)    The metrics in Table 3 are not a standard set of metrics for saliency prediction. Why don’t the authors report the complete of evaluation metrics in Table 3? It seems strange to drop the popular metrics such as sAUC and SIM in Table 3. It is a standard practice to report all standard metrics, ie., those commonly used in saliency predictions, as in the online benchmark (https://saliency.tuebingen.ai/evaluation.html). It seems odd to me the authors only report selected metrics in Table 3 but report all in Table 6 (which is in the appendix).

b)	Following a), why do the authors put both Table 3 and Table 6? Both of the tables occupy the same space. The authors can just put Table 6 in the main manuscript instead of Table 3.

c)	Why do the comparison methods vary greatly across different testing datasets? It will be helpful if the authors stick to several methods and compare their performance across the same testing datasets. Deep Gaze II and IIE are recognized for their strong performance on images of general scenes. Why did the authors choose to test these models only on the CAT2000 dataset and not on other general scene datasets such as Salicon and OSIE?

d)	In the SalGAN paper [60], it reports the metrics on Salicon dataset, why the authors do not report this in Table 3 or Table 6 for SalGAN on Salicon dataset, but instead, only report its performance on the webpage dataset? If the authors have simply copied results from other works, incorporating these additional benchmarks would require minimal effort and should have been included. If the authors have tested the models, why not test them on the popular benchmarks such as Salicon and OSIE?

e) Can the authors explain why they don’t report SequenceScore for COCO-Search18? This column is denoted as "-" in Table 4.

f) Furthermore, the reported results on COCO-Search18 of the comparison method FFM[79] in its own paper [79] are different from what what authors report in Table 4, both under target present conditions. The SemSS scores in [79] under target present conditions are consistenly above 0.53, and the method of Chen et al. [16] is 0.572. Can authors provide more details on how they obtained the current results in Table 4 for the comparison methods?

g) Can authors explain why they didn’t compare scanpath prediction performance of ALOHA and baselines on COCO-FreeView in Table 4 even though they use COCO-FreeView in Sec. 4.4?

h)	It will be helpful if the authors can experiment with more popular benchmark datasets such as MIT300 and MIT1003, and compare with the performances published on the online benchmark (https://saliency.tuebingen.ai/results.html).

i)     Minor: some top performances are wrongly indicated, eg., on the webpage dataset (FiWI), the best performance on KLD should be Chen et al. [13].


In summary, excluding the SOTA models in common benchmark datasets significantly undermines the experimental evidence for ALOHA's competitiveness, especially considering that 7 out of the 11 datasets in the experimental section are for free-viewing saliency prediction (see Table 1). Based on all the above, I feel that current experiments cannot adequately supported the paper's claims.The paper needs more discussions and comparison results to support its claim that the model achieves SOTA performance.

2.	Although the authors claim that the unified model could “act as a comprehensive reward model, predicting subjective ratings/preferences as rewards and providing additional insights from predicted human attention and behavior patterns,” the experiments do not clearly demonstrate the actual benefits of unifying the attention, rating, and scanpath tasks. The motiviation of this work is not well positionied.

3.	Some technical details of the paper are not clear. E.g, a). the three predictors appear to operate independently, but it is unclear how they were chosen. b) Why text generation is chosen for scanpath generation. c) How is the model fine-tuned on various datasets.

**Questions:**

1. How are the three predictors chosen for each task? Are they fixed assigned based on different input texts?
2. Are there any differences between the salience map and the importance map in terms of architecture or loss function, or do the differences lie solely in the training data?
3. In the "decoding during inference" section, the paper mentions, “If there is no fixation available in the predicted string, we mark the scanpath as invalid.” This invalid case (“no fixation”) only covers situations where all token pairs are non-numerical. If some tokens in the output sequence are in an unrecognizable format, rendering part of the results invalid, how should this be evaluated and handled?
4. Why is text generation used as a scanpath predictor?
5. How is the model fine-tuned on various datasets, which parts are freezed?
6. All my above questions regarding the numerical results (Table 3 and Table 6) in the Weakness section.

**Limitations:**

The reported numerical results do not adequatly support the paper's claim.

More experiments to demonstrate the model's task transferring ability are needed. For example, transferring from saliency map to rating, or from scanpath to saliency map.

---

> ### Author Rebuttal · Authors · 2024-08-07
>
> Thank you for the comments and we address each point below.
>
> ---
>
> **Metrics in Table 3 and Table 6**
>
> A: We include the important metrics and baseline methods in Table 3 so that it is more readable with larger font size. We have included reference to Table 6 in Table 3 caption for completeness. Some metrics are not chosen in Table 3, because the baseline papers did not report them, and hence caused empty columns in Table 6.
>
> ---
>
> **Different comparison methods across different testing datasets**
>
> A: For fair comparison, we refer to each baseline’s performance in their original papers and follow the same evaluation protocols / metrics, which can vary on different benchmarks.
>
> ---
>
> **Why test Deep Gaze II / IIE only on CAT2000 and not on Salicon and OSIE?**
>
> A: Deep Gaze II / IIE papers reported results on CAT2000 but not on Salicon validation set / OSIE. We did not re-implement any baseline due to the large number of datasets / tasks we needed to compare.
>
> ---
>
> **Why not include SalGAN on the Salicon dataset in Table 3 or Table 6?**
>
> A: There are two versions of Salicon data, Salicon 2017 (http://salicon.net/challenge-2017) and 2015 http://salicon.net/challenge-2015). All results in our table are on the newer Salicon 2017 dataset, in line with the baselines in Table 3, while SalGAN results were obtained on the Salicon 2015 dataset. The 2015 version differs in fixation ground truth, affecting metrics like NSS scores (higher for Salicon 2015 than 2017), so the results on Salicon 2015 are not comparable to Salicon 2017.
>
> ---
>
> **Why no SequenceScore for COCO-Search18?**
>
> A: The computation of the SequenceScore requires a standardized superpixel segmentation, which is not available after we communicated with the authors. To avoid inaccurate / unfair comparison, we did not include this metric on this dataset.
>
> ---
>
> **FFM [79] results on COCO-Search18 different from the ones in Table 4?**
>
> A: We refer to the results of FFM in Gazeformer [56] paper, which appeared in Table 6 in Appendix. There were some issues regarding the evaluation results on COCO-Search18 due to the misalignment of the ground-truth semantic segmentation map. The authors have updated their results in Appendix, which are the correct version that we cited in Table 4.
>
> ---
>
> **Why not compare scanpath prediction performance of ALOHA and baselines on COCO-FreeView?**
>
> A: We have the scanpath results on COCO-FreeView, but as a relatively new dataset, we did not find a good baseline for scanpath prediction on this dataset, so we did not include this dataset in our table. One relevant work is [1]. However, it didn’t report scanpath prediction performance on COCO-FreeView.
>
> [1] Characterizing target-absent human attention. (CVPR'22)
>
> ---
>
> **Additional experiments on MIT300 and MIT1003**
>
> A: We have a model that did not include MIT1003 as training and directly tested on MIT300, and the results are suboptimal. However, since many methods on MIT300 benchmark dashboard are trained on MIT1003, so for a fair comparison, we will train a model on MIT1003 and report the results on MIT300.
>
> ---
>
> **The experiments do not clearly demonstrate the actual benefits of unifying the attention, rating, and scanpath tasks**
>
> A: The immediate benefit of an unified model is that only one model is needed instead of separate models for each task, leading to easier model serving with better performance. Furthermore, there are strong connections between attention and subjective experiences like aesthetic preference, subjective interpretation, emotional response, or ease of information finding (e.g., often focused attention means easy, while scattered attention means hard). To this end, we collect and will release a new dataset which contains both scores (easiness scores for question answering tasks) and gaze heatmaps (while performing question answering tasks) on digital images (designs, UIs, etc) , and show that the score and entropy of gaze heatmaps are correlated,  to better motivate the unified model. Please see more details in Figure 3 and 4 in the rebuttal PDF.
>
> ---
>
> **How are the three predictors chosen for each task?**
>
> A: During training, we specify the task in the text prompt and compute gradients only using the loss of that task (e.g., scoring task if “OUTPUT: score” is in the prompt). During inference, the task info (i.e., “OUTPUT: score”) in the prompt will inform the model which task to perform and we retrieve results from the corresponding predictor.
>
> ---
>
> **Differences between the salience map and the importance map?**
>
> A: The two tasks use the same heatmap L2 loss and only differ in data. Note that the two tasks will have different text prompts, namely “OUTPUT: saliency heatmap” for saliency, and “OUTPUT: importance heatmap” for importance.
>
> ---
>
> **Invalid output format for scanpath prediction?**
>
> A: We first split the output sequence by the separator (“and” in our case). Then each component should contain two coordinates of x and y. If a <x, y> coordinate is invalid (e.g., only x but no y value), we skip that coordinate. All predicted scanpaths are valid after the models were finetuned.
>
> ---
>
> **Why text generation as a scanpath predictor?**
>
> A: Text decoder can generate arbitrary length of tokens (up to max length) and is suitable for scanpath which also has variable length. As the vision-language model is pretrained on coordinate prediction task the decoder learned the notion of coordinate token and performed well for predicting scanpath, which mainly consists of coordinates. As a unified model, we hope the model can generalize to different kinds of sequences, which can be predicted by the text decoder.
>
> ---
>
> **How is the model fine-tuned on various datasets, which parts are freezed?**
>
> A: All model weights are finetuned and no parts are frozen. We randomly sampled batches from all the datasets during training (more details in Section 4.2).

---

> > ### Comment · Reviewer_Fbj6 · 2024-08-09
> >
> > Thank you to the authors for the rebuttal. However, I remain unconvinced by the explanations provided. As a researcher in the saliency field, I consider sAUC and SIM to be essential metrics for evaluating saliency prediction, yet they are notably absent from Table 3. This omission challenges the authors' assertion that Table 3 includes "the important metrics."
> >
> > Furthermore, in a scientific research paper, it is crucial to include key metrics and provide comprehensive evaluation, which should take precedence over concerns about font size. Therefore, I find the decision to place detailed information in Table 6 in the supplementary material, while presenting only partial and duplicate information in Table 3 in the main paper, to be unjustified, especially when both tables occupy the same length. The authors' reasoning to make Table 3 "more readable with a larger font size" seems insufficient and unconvincing.
> >
> > Similarly, the explanation that the authors "did not find a good baseline" does not sufficiently justify the exclusion of scanpath results for COCO-FreeView, particularly given that COCO-FreeView is used in Section 4.4. Moreover, the authors did not adequately address the question regarding the SemSS scores in [79], where scores under target-present conditions are consistently above 0.53, and the method of Chen et al. [16] achieves 0.572.
> >
> > The examples provided to demonstrate the benefits of unifying attention, rating, and scanpath tasks are rather weak and not sufficiently convincing. First, the examples involve only attention and rating, without scanpath, which undermines the claim of presenting a truly unifying example. Second, the observed correlation between score and heatmap entropy is expected, as it is natural for more challenging tasks to result in more complex browsing patterns. Additionally, the correlation observed is weak by statistical standards.
> >
> > In view of the above, I would like to keep my rating as reject.

---

> > > ### Author Response · Authors · 2024-08-12
> > > **Response**
> > >
> > > Thank you for your prompt reply, we really appreciate it! We would love to have another round of response for some clarifications, and hope those help!
> > >
> > > ---
> > >
> > > **Table 3 & Table 6**
> > >
> > > We agree that there are some duplications between these two tables. Our intention in including Table 3 was primarily for presentational purposes.
> > >
> > > We are happy to replace Table 3 with Table 6 in the main paper to include metrics sAUC and SIM, or the full set of eight metrics.
> > >
> > > ---
> > >
> > > **Scanpath results for COCO-FreeView**
> > >
> > > As one of the dataset we used for the experiment, we have the results from our model on COCO-FreeView. We would welcome the opportunity to compare our results with established baselines. If you could kindly direct us to a publicly available benchmark for scanpaths on the COCO-FreeView dataset, we would be glad to include a comparison with these baselines in our paper.
> > >
> > > ---
> > >
> > > **SemSS score from FFM [79]**
> > >
> > > Regarding the discrepancy in results, we have taken steps to clarify the issue.
> > >
> > > We contacted the first author of FFM [79] directly to discuss the mismatch. The author confirmed that the results of this experiment in FFM [79] are indeed problematic due to incorrect labels. With the author's permission, we are willing to share their response for your reference, which validates our approach of referring to the updated results in Gazeformer [56].
> > >
> > > > The SemSS of FFM in the two papers (GazeFormer and FFM) are different, because there was a mistake in generating the semantic labels in the FFM paper. Labels got mismatched in image size, which leads to systematic shift in SemSS for all methods (namely, all methods got higher SemSS scores, but the order maintains). We should refer to Table 6 (b) in GazeFormer as the more reliable SemSS scores for GazeFormer, FFM, and Chen et al.
> > >
> > > Hope this clarifies the question.
> > >
> > > ---
> > >
> > > **Unifying attention, rating, and scanpath**
> > >
> > > 1. Scanpath is naturally correlated with attention map: an attention map is aggregated from scanpaths from multiple users. In this way, rating, attention, and scanpath could be unified via a prediction model. We do have scanpath data, however, we will be cautious about releasing the scanpath data, due to privacy concerns.
> > > 2. We are happy that you agree the observed correlation between score and heatmap entropy is expected. This verified evidence further motivates the unified model, which is the foundation for studying the correlation between the heatmap and score for better performance. There are many other cases where subjective ratings/experiences and human attention (heatmap) are related, e.g., “Salient-Centeredness and Saliency Size in Computational Aesthetics”, ACM Transactions on Applied Perception 2023; “Aesthetic Attention”, https://philpapers.org/archive/NANAA-2.pdf. So a unified model to predict subjective ratings/experiences and human attention/scanpath will help advance the research in this direction.
> > > 3. We observed a p-value around 7e-7 for the correlation between easiness score and heatmap entropy, indicating the correlation between easiness score and gaze heatmap entropy is statistically significant.
> > >
> > > ---
> > >
> > > We hope our clarification and the follow-up response will help us better understand our work. A reconsideration of our work is highly appreciated.

---

### Official Review · Reviewer_ew4J · 2024-07-11

**Soundness:** 4
**Presentation:** 4
**Contribution:** 4
**Rating:** 7
**Confidence:** 4

**Summary:**

Noticing the existing issues in human behavior modeling, such as isolating the study of implicit, early-stage perceptual behavior (like human attention) from explicit, later-stage behavior (like subjective preferences), specified visual content type; in this manuscript, the author(s) aimed to build an integrated human attention and preference behavior model for addressing multiple visual content types. Through empirical experiments, the author(s) demonstrated the effectiveness of the proposed model.

**Strengths:**

In my opinion, the strengths of this manuscript are as follows:

1. Designed a unified approach to model human visual behavior: image+text=>human perceptual behaviors.

2. Extensive experiments are performed to validate the effectiveness of the proposed model.

**Weaknesses:**

In my opinion, the weaknesses of this manuscript are as follows:

1. This manuscript seems not to discuss the model's limitations in real-world situations, such as how its performance is in dynamic environments or its adaptability to changes in user behavior over time.

2. As the author(s) described, ALOHA has 848 million parameters. Training such a large model requires significant computational resources, which may not be accessible to most researchers or practitioners.

**Questions:**

I read the manuscript, and I have the following questions/comments. Thanks.

1. Are there any potential applications of ALOHA in real-world scenarios or in dynamic environments? How can it be used to optimize elements, such as user interfaces, graphic designs, and content creation?

2. Are there any potential biases in the current model's predictions?

Tiny format issues in References:

(1) Sometimes, the journal/conference name used an abbreviation; sometimes, not, sometimes, both, such as Ref.[25], Ref.[67].

(2) Keep the format of the references consistent, such as Ref. [6] vs. Ref.[15].

Please check carefully and correct the issues.


Overall, ALOHA represents a significant advancement in modeling human visual behavior, offering a unified approach that spans from early-stage perceptual responses to later-stage decision-making. I think this is an interesting manuscript.

I look forward to hearing from the author(s). Thanks.

**Limitations:**

Yes.

---

> ### Author Rebuttal · Authors · 2024-08-07
>
> Thank you for the comments and we address each point below.
>
> ---
>
> **Model's limitations in real-world situations, such as how its performance is in dynamic environments or its adaptability to changes in user behavior over time.**
>
> A: Current model does not consider dynamic environments or changes in user behavior over time, which is indeed a limitation. We will add this limitation to the discussion section. Please also see our response to the question of “Stay up to date” from reviewer FgZM.
>
> ---
>
> **ALOHA has 848 million parameters. Training such a large model requires significant computational resources, which may not be accessible to most researchers or practitioners.**
>
> A: We acknowledge the resource requirement might be difficult for researchers without access to enough GPUs / TPUs. There are techniques to reduce model training cost, like LoRA, or Parameter-Efficient Fine-tuning (https://huggingface.co/blog/peft), which could be employed with much less resources. However, using such techniques would be beyond the scope of this work. We will include this info in the paper and develop models with lower resource requirements in our future work.
>
> ---
>
> **How can it be used to optimize elements, such as user interfaces, graphic designs, and content creation?**
>
> A: ALOHA can help optimize content in several ways. For example, the predicted saliency heatmap can be used to remove the distracting areas in visual content, similar to “Deep Saliency Prior for Reducing Visual Distraction”:  https://arxiv.org/abs/2109.01980. If the content is generated by a generative model, ALOHA’s predicted score can be used as a reward score to improve the generative model, with learning from human feedback/preference method, such as DPOK (https://arxiv.org/pdf/2305.16381) and DRaFT (https://arxiv.org/abs/2309.17400).
>
> ---
>
> **Are there any potential biases in the current model's predictions?**
>
> A: Please see the global rebuttal.
>
> ---
>
> **Format issues**
>
> A: We will make the edits according to the feedback.

---

### Official Review · Reviewer_FgZM · 2024-07-11

**Soundness:** 3
**Presentation:** 3
**Contribution:** 3
**Rating:** 5
**Confidence:** 3

**Summary:**

This paper introduces ALOHA, a multimodal model that predicts human saliency, scanpath, and subjective rating of natural images, webpages, and graphic designs. ALOHA outperforms or performs similarly to baseline models across each of its prediction tasks while improving generalizability over task-specific models.

**Strengths:**

*Generalizable model of human visual behavior and preferences.* The ALOHA model predicts multiple forms of human preference --- the saliency, scanpath, and rating --- of various types of image inputs, including web pages, natural images, and cartoons. This improves the generalizability of ALHOA over prior models that focus on a single prediction task or data input modality.

*In-depth analysis of model performance compared to prior benchmarks.* The paper provides a thorough evaluation of ALOHA's task-performance across 13 metrics, 11 datasets, and 4 tasks. ALOHA outperforms or performs on par with pre-existing models across all evaluations.

*Easy to read paper.* The paper is well motivated and clearly explained. Figure 1 is a particularly helpful overview.

**Weaknesses:**

My biggest concern is the paper’s lack of engagement with the limitations and ethical considerations of modeling subjective human preferences. Currently there is a brief discussion of training data limitations in the supplement. I would like to see a thorough discussion of considerations included (or at least referenced) in the main text. Possible discussion points that come to mind include:

* What are the ethical considerations of replacing humans with models trained to replicate their preferences? Humans have diverse and often opposing preferences, particularly for subjective notions of aesthetics or attention. It is possible that models trained to replicate human preferences could learn a more uniform or singular notion of preference. Could using these models limit the visual diversity in webpages and content creation? Or could these models actually help us generate more visually diverse content?
* Whose preferences are included in the model and, more importantly, whose aren’t? Humans likely have diverse preferences when it comes to subjective notions like website attractiveness. I would suspect these preferences vary based on age, experience with technology, culture, etc. Building a model to replicate human perspectives may amplify a particular worldview that is not representative of all users. On the other hand, it may be easier to create a model trained on diverse human data than to get a diverse set of human feedback for every new webpage design.
* The paper suggests using the model as a reward function to optimize content creation. Could this result in negative consequences, such as content that is optimized for a model’s preference function and may not be value aligned with what humans want?
* How could human preference models, like ALOHA, be misused? For example, it seems like these models could be used to make ads more intrusive by optimizing their scanpath location or to make phising sites more convincing by optimizing the saliency of the credential input.
* How should human preference models take into account blind or low-vision users? Scanpaths and saliency likely differ between users who rely on screenreaders and those who don’t. Optimizing the web for sighted preferences could further exclude blind and low-vision from online spaces. Given the [emphasis](https://www.ada.gov/resources/web-guidance/) on web accessibility, how can we use human preference models to amplify (not weaken) online accessibility?
* How should models stay up to date with the changing human preferences? Human preferences, especially around aesthetics, are constantly changing. As a result, what are the implications for interpreting the evaluations used in this paper that span datasets from 2014–2023?

The paper would also be strengthened by an audit of their model’s behavior beyond quantitative performance metrics, such as an analysis of the diversity of the humans included in the training dataset, a categorization of the types of mistakes the model makes, and a [Model Card](https://arxiv.org/abs/1810.03993) reflecting its design.

I recognize that the paper is inheriting these ethical concerns from the preexisting datasets and tasks that it works on. I hope that by including a discussion of the models implications, the paper will progress research into these types of models and inspire work on new datasets, model analysis, and safe deployment techniques.

I might also suggest that the authors should select a new name for their model. I am not an expert in Hawaiian culture, but I know that the term ‘aloha’ is culturally significant, and there have been [Hawaiian movements to reclaim it](https://www.usatoday.com/story/life/health-wellness/2023/01/13/stop-saying-aloha-out-of-context/10990192002/), such as the [‘Aloha Not For Sale’ protests](https://kawaiola.news/cover/aloha-not-for-sale-cultural-in-appropriation/). Using it out of context as a model name, may minimize the rich cultural meaning behind the term, and given it is an imperfect acronym anyway, should be an easy change.

**Questions:**

Please see the Weaknesses section above.

**Limitations:**

Please see the Weaknesses section above.

---

> ### Author Rebuttal · Authors · 2024-08-07
>
> We appreciate you bringing the ethical concerns to our attention. We recognize that such concerns are common with machine learning models, particularly those involving user preference and behavior modeling. We are committed to addressing these issues to the best of our ability by expanding the ethics and limitations section.
>
> ---
>
> **Uniform vs diverse human preferences**
>
> We acknowledge that models carry the risk of converging towards a more uniform notion of preference, a concern shared by all machine learning models. To promote visual diversity in our model, we propose,
> 1. Hybrid Approach: Initially, the model should be used in a hybrid manner, providing insights without replacing human decisions in web optimization, and
> 2. Personalized Models: Develop personalized models based on our initial unified model. This approach will help generate more diverse predictions based on user attributes.
>
> ---
>
> **Demographics of annotators**
>
> Please see our response in the global comment.
>
> ---
>
> **Aligning with human preference**
>
> While our model should be approximately aligned with human preferences, we recognize that using it as a reward model may lead to reward hacking. We plan to incorporate techniques [1, 2] for mitigating this when we use it as a reward mode.
>
> [1] Parrot: Pareto-optimal Multi-Reward Reinforcement Learning Framework for Text-to-Image Generation
> [2] ReNO: Enhancing One-step Text-to-Image Models through Reward-based Noise Optimization
>
> ---
>
> **Properly using ALOHA model**
>
> Like many technologies, human gaze prediction models could potentially be misused. Without regulations, such as unrestricted access to users without credentials and irresponsible data collection, the risk can be increased. To ensure the proper usage of our model, we propose,
> 1. Placing restrictions on model access to prevent unauthorized applications or users;
> 2. Adhering to strict ethical guidelines on data collection, and actively monitoring the collection process at scale; and
> 3. Being transparent about our model's capabilities and limitations. We believe that keeping humans in the loop and only using the model prediction as a reference but not replacing humans is important.
>
> Internally, we strive to follow the ethical guidelines of our institution, ensuring that the model usage remains controlled and responsible. This has also caused us to be cautious about making our model publicly available.
>
> ---
>
> **Blind and low-vision users**
>
> Our model can benefit blind and low-vision users for screen users by highlighting the most important areas of a webpage via heatmap predictions and making them more accessible via screen readers. We recognize that the current training data may have limitations in representing the full spectrum of user experiences. We plan to enhance the inclusiveness by,
> 1. Multi-Modal Preference Modeling: We're developing our model to incorporate not just visual cues, but also how users interact with content through screen readers, voice commands, and other assistive technologies;
> 2. Collaboration with Accessibility Experts: We plan to collaborate with accessibility experts and organizations representing blind and low-vision users for future iterations of our work.
>
> ---
>
> **Stay up to date**
>
> Staying up to date is essential for models, and can be achieved by fine-tuning with more recent data. Continual learning techniques also support this idea [1]. However, updating the training data is out of the scope of our paper, as we want to focus on developing the first unified model on the diverse user modeling tasks.
>
> [1] A comprehensive survey of continual learning: theory, method and application
>
> ---
>
> **Error Analysis**
>
> We analyzed 50 error examples in the Koniq-10K dev set and found some interesting patterns, see examples in Figure 1 in our rebuttal PDF. The first example shows the most common error category when our model predicts a higher score for colorful but blurry images. The second example shows another common model error where a low-light image gets a higher prediction score. For the last black-and-white example, on the contrary, our model predicts a lower score than the groundtruth. These errors demonstrate that human preferences for image aesthetics can depend on diverse factors including clear focus, correct lighting, and artistic style.
>
> Similarly, for heatmap prediction, we analyzed the 30 examples with the lowest NSS scores on the Salicon validation set. The example in Figure 2 in the rebuttal PDF demonstrates the most common error category where the groundtruth contains a more scattered gaze heatmap. In such cases, our model prediction might not focus on the right objects.
>
> We will conduct more error analysis on other model predictions and include this info in the paper.
>
> ---
>
> **New content to include**
>
> Happy to incorporate your recommendations and include a discussion on the following action items:
> 1. The diversity / demographics of annotators
> 2. Model failure cases and discussions
> 3. A model card
>
> ---
>
> **Model’s name**
>
> We are happy to choose a new name from alternative names for our model, for example,
> 1. GLAM: From Glances to Likes: A Unified Model for Understanding Human Visual Behavior
> 2. TOTORO: from attention TO likes TO human RespOnses – a unified model of human visual behavior
> 3. HABITAT: modeling Human Attention and Behavioral InTeractions Across diverse visual content using a multimodal Transformer

---

> > ### Comment · Reviewer_FgZM · 2024-08-08
> >
> > Thank you for taking the time to think deeply about the implications of this research! I appreciate your diligence in finding demographic information and doing error analysis, your thoughtfulness in responding to each of my concerns, and your commitment to include a model card and these considerations in the paper.
> >
> > In addition to what you have listed in "New content to include", I would suggest:
> > * Beyond discussion limitations of the demographic information that exists, I suggest also discussing limitations in the demographic information that is not reported. From your demographic analysis, we only know the annotators age and gender. However, we do not know their location, race, country of origin, education level, etc. --- all of which contribute to someone's worldview and could change the way they interact with a webpage.
> > * I would like to your discussion section include all of the limitations you have mentioned in this rebuttal --- a uniform model for diverse human preferences, reward hacking, safety measures for human preference models, making these models accessibility friendly, etc --- not just the demographic info, error analysis, and model card.
> >
> > I appreciate your suggestions for new names. I think all the ones you suggested are great, so I will leave the name decision up to you.
> >
> > Assuming these changes are reflected in the final manuscript, I am happy to accept the paper and have increased my score accordingly.

---

> > > ### Author Response · Authors · 2024-08-09
> > > **Reply**
> > >
> > > Thank you for helping us improve our paper and increasing your rating!
> > >
> > > > discussing limitations in the demographic information that is not reported
> > >
> > > That's a very good idea and thanks for your kind reminder! We will make sure to involve this part in our final discussion on demographic information. By quoting all the available information we have, we will also discuss the missing information in demographics, which could be useful for future reference.
> > >
> > > > include all of the limitations you have mentioned in this rebuttal
> > >
> > > For sure, we will add these thoughts in our rebuttal to the final version of the paper with a clear structure.
> > >
> > > We appreciate your effort in reviewing this paper and we really enjoy talking to you.

---

### Official Review · Reviewer_RXVY · 2024-07-13

**Soundness:** 3
**Presentation:** 3
**Contribution:** 3
**Rating:** 5
**Confidence:** 4

**Summary:**

In their paper "ALOHA: from Attention to Likes – a unified mOdel for understanding HumAn responses to diverse visual content" the authors describe a new unified model to predict human saliency(attention/importance), even more fine grained than that, scanpath, and ratings.

After nicely introduced the motivation and an extensive outlook on related work, the authors describe the multimodal VLM-based encoder-decoder transformer architecture in detail, especially the three predictor head. Exploting the power of instruction tuned LLM, the authors introduce additional tokens to  predict valid scanpath at inference time.

The benefits of their model are evlauted based on many experiments, which are shortly discussed, followed by a short conclusion.

**Strengths:**

The main strength of the paper are the experiments, which are extensive across several benchmarks and metrics.
Integrating a three headed VLM for the tasks is intuitive and elegant.
Their proposed ALOHA architecture is SOTA in sevaral settings (22/35)
The writing is mostly easy to follow and there is a clear picture the authors want to paint.

**Weaknesses:**

- The writing can be improved in several places, I'll note here a few:
1. Introduction:
- In the introduction, it seems that the authors already have the conclusion at the end, switching from present to past tense for the main contributions of the paper. This makes it sounds as if the authors already published ALOHA previously.
- In the same two points of the main contributions, the authors actually only have one contribution, their model ALOHA. The second "contribution" is only the evaluation of ALOHA.
- regarding the previous comment -- the second contribution, even though from my point of view it is no contribution is the evaluation of ALOHA and not the training of ALOHA.

3. Unifying Human Visual Behavior Modeling from Attention to Likes:
- The overall optimization criterion should be part of the main paper, not in the appendix.
- Model training (section 4) from my pov should be part of section 3, it is not part of the experiments since you do not report multiple training strategies.

Its unclear how hyperparameters were tuned.
Code will not be made public, which is a major concern in the current reproducibility crisis in ML. Additionally the question in the questionnaire is wrongfully answered as N/A; the paper DOES INCLUDE experiments requiring code. (GUIDELINE: The answer NA means that paper does not include experiments requiring code.)

**Questions:**

You mention that you pad images to 512x512 -- what happens with images larger than 512x512?

How did you arrive at your optimization criterion, scaling for the learning rates and in general all hyperparameters? You did not report any hyperparameter tuning.

**Limitations:**

Yes.

---

> ### Author Rebuttal · Authors · 2024-08-07
>
> Thank you for the comments and we address each point below.
>
> ---
>
> **The writing can be improved in several places**
>
> A: We will edit the paper according to the suggestions.
>
> ---
>
> **How hyperparameters are tuned**
>
> A: We did a training-validation split on our larger datasets and then tuned the hyperparameters (e.g., learning rate, batch size, dropout rate, loss weights of the 3 heads) on the validation set through grid search. The model was then trained with the full training data with the chosen hyperparameters.
>
> ---
>
> **What happens with images larger than 512x512?**
>
> A: We resize the images to have a max height or width of 512 and center-pad the smaller dimension to 512.
>
> ---
>
> **Code will not be made public, which is a major concern in the current reproducibility crisis in ML**
>
> A: Given the sensitivity of this topic, and our concern that the code may be misused if made public (see comments from Reviewer FgZM and our rebuttal on "Properly using ALOHA model"), our institution has a strict policy on opening source code in this area. But to help advance the research, we will provide enough details (including all the info in the rebuttal), all necessary communication and support for o researchers in this area to reproduce our work, who we believe will make use of this technique appropriately.

---

### Author Rebuttal · Authors · 2024-08-07

We thank the reviewers for their thorough and constructive feedback. We have addressed each point in the individual responses. We have included some of the common points of discussion as below. Moreover, in the rebuttal pdf, we also included some figures to answer the questions of “Error Analysis” from reviewer FgZM, and “the actual benefits of unifying the attention, rating, and scanpath tasks” from reviewer Fbj6. We will also fix all the formatting issues pointed out by the feedback.

---

**Demographics of annotators [Reviewers FgZM, ew4]**

We check the training data collection processes and include the participant demographics that we have found.

WS-Saliency: “A total of 41 participants (19 females, 22 males; age range 17-23; with normal or corrected-to-normal vision) participated in our data collection.”

Mobile UI: “Thirty participants (12 male, 18 female). [...] The average age was 25.9 (SD=3.95). The participants had normal vision (8) or corrected-to-normal-vision (22). Twenty of the 22 wore glasses and the remaining two wore contact lenses.”

Imp1k: “The data of 43 participants (29 male, most in their 20s and 30s) were used in the resulting analyses.”

FiWI: “11 students (4 males and 7 females) in the age range of 21 to 25 participated in data collection. All participants had normal vision or corrective visual apparatus.”

Based on the available descriptions, we found that participants show good convergence on gender. However, the age distribution is somewhat skewed, likely because most collections were conducted at universities. Datasets like Salicon and Koniq-10K, which utilize crowdsourcing workers for annotations, are expected to have a better balance in terms of age and other attributes. We will add this information to the limitations discussion.

---

**Bias in the data sets or the model [Reviewer ew4]**

The model is trained with multiple public data sets, each of which might have some bias. So it is possible for our model to have also learned such bias from those datasets. Since this paper introduces the first model to unify all the human visual behavior tasks, we focus on getting a model to work for these tasks with good accuracy, and we plan to implement techniques to evaluate and mitigate bias in future iterations of our work. We will however add this discussion in our limitation section along with demographics information discussed above.

---

We look forward to the discussion phase and further improving the paper.

---

### Decision · Program_Chairs · 2024-09-25

**Decision:**

Accept (poster)

**Comment:**

The paper received 2 Borderline Accept, 1 Accept and 1 Reject ratings. While the authors could incorporate more experimental validations, the paper is well motivated and makes a solid contribution in unifying attention prediction. The AC reads the paper, the rebuttal, and the discussions and decides to accept it. Congratulations! It is encouraged that the authors incorporate reviewers' comments into their final version.